# Machine learning reveals distinct T-cell receptor clusters in plasma cell dyscrasias compared to healthy controls

David G. Coffey[ID][1]*[◕], Yong Zhang[2◕], Elizabeth Hill[3], Frank Cross Jr.[2], Reena Philip[4], Marc R. Theoret[2,4], Ola Landgren[1], Andrea C. Baines[ID][2]*, Dickran Kazandjian[1]

**1** Division of Myeloma, Sylvester Comprehensive Cancer Center, University of Miami, Miami, Florida, United States of America, **2** Center for Drug Evaluation and Research, US Food and Drug Administration, Silver Spring, Maryland, United States of America, **3** Lymphoid Malignancies Branch, National Cancer Institute, National Institutes of Health, Bethesda, Maryland, United States of America, **4** Oncology Center of Excellence, US Food and Drug Administration, Silver Spring, Maryland, United States of America

◕ These authors contributed equally to this work.
* davidcoffey@miami.edu (DGC); Andrea.Baines@fda.hhs.gov (AB)

## Abstract

T-cell receptor (TCR) repertoire diversity has been implicated in the progression and prognosis of multiple myeloma (MM). This study aimed to evaluate the association between T-cell clonality, immune response, and clinical outcomes in patients with plasma cell dyscrasias using next-generation sequencing of the TCR β chain (TCRB). TCRB sequencing was performed on peripheral blood samples from patients with monoclonal gammopathy of undetermined significance (MGUS), smoldering multiple myeloma (SMM), and newly diagnosed multiple myeloma (MM). Healthy individuals served as controls. No significant differences in TCR repertoire diversity were observed between healthy individuals and those with MGUS, SMM or MM after adjusting for age. Furthermore, TCR diversity did not correlate with treatment response in newly diagnosed MM or SMM patients. However, machine learning analysis revealed distinct TCR clusters differentially abundant between healthy individuals and those with plasma cell dyscrasias, exhibiting different amino acid properties. These findings suggest that shared T-cell receptor specificities among patients with plasma cell dyscrasias reflect underlying differences in antigen recognition and underscore the need for further studies to unravel the functional and clinical significance of these distinct immune signatures.

## Background

Multiple myeloma (MM) is a plasma cell neoplasm consistently preceded by the asymptomatic conditions of monoclonal gammopathy of undetermined significance (MGUS) and smoldering multiple myeloma (SMM). The complex interplay between the malignant clone and the host immune system plays a pivotal role in disease

**Data availability statement:** The TCRB sequencing data generated in this study as well as the previously data from healthy individual are available through the ImmuneACCESS data portal: https://clients.adaptivebiotech.com/pub/coffey-2025-s (DOI 10.21417/DGC2025S).

**Funding:** The author(s) received no specific funding for this work.

**Competing interests:** The authors have declared that no competing interests exist.

progression and response to therapy. T cells, as key components of the adaptive immune response, have been extensively investigated in MM [1].

Each T cell expresses a unique T-cell receptor (TCR), most commonly composed of an α and a β chain, that determines antigen specificity through its interaction with peptide–MHC complexes. The complementarity-determining region 3 (CDR3) of the TCR, particularly within the β chain, is the most variable and critical for recognizing antigenic peptides [2]. TCR diversity refers to the number and relative distribution of unique CDR3 sequences within a repertoire, capturing both the richness of distinct clonotypes and their evenness of representation. In MM and its precursor conditions, decreased repertoire diversity may indicate antigen-driven clonal expansions or T-cell senescence, whereas preserved diversity may reflect intact immune surveillance. Repertoire analysis can therefore provide important insights into immune competence, disease progression, and treatment response.

Prior research using low-resolution approaches, such as flow cytometry with anti-TCR variable-β monoclonal antibody panels, identified clonal expansions of T cells in both the peripheral blood and bone marrow of patients with MM [3,4] The phenotype of these expanded T cell clones has predominantly been reported to be long-lived CD8+ CD28- T cells [5]. The absence of CD28 expression, a T cell costimulatory molecule expressed by naive T cells and lost during maturation into activated and effector cytotoxic T lymphocytes, indicates chronic antigen stimulation [6]. This clonal expansion has been suggested to represent a host immune response against an undefined antigen stimulus. However, the exact nature of this antigen remains elusive, as studies using tetramers to identify the specificity of the expanded T cell clones have shown no specific reactivity towards CMV, MM idiotypes, or other autologous antigens [7].

Notably, the occurrence of clonal T cell expansion has been linked to better progression-free survival (PFS) and overall survival (OS) among MM patients [8], particularly in those presenting with a lower tumor burden [9]. This association suggests a potential anti-tumor effect of the expanded T cell. Nevertheless, with the introduction of next-generation sequencing (NGS) technologies, which provide far greater resolution in analyzing the TCR repertoire, these survival associations have not been consistently observed. For instance, our recent study comparing the peripheral blood T cell repertoire diversity using NGS did not find a significant difference in patients achieving sustained versus not sustained minimal residual disease (MRD) negativity during lenalidomide maintenance [10].

This study was designed to reassess the T cell repertoire in patients with plasma cell dyscrasia by utilizing NGS techniques, with the goal of exploring the relationships between T cell diversity and clinical outcomes. By leveraging the enhanced resolution and sensitivity of NGS, we seek to provide a more comprehensive understanding of the T cell repertoire in plasma cell dyscrasias and its potential implications for disease prognosis and therapeutic strategies.

## Methods

### Study population

This study was approved by the National Cancer Institute Institutional Review Board. Written informed consent was obtained from all participants in accordance with the

Declaration of Helsinki Protocol and the International Conference on Harmonization. TCR sequencing was performed on the peripheral blood mononuclear cells (PBMCs) of 80 patients with untreated MGUS, 55 with SMM, and 31 with newly diagnosed MM before treatment (NDMM, S1 Table).

Patients with SMM were enrolled in a phase 2 nonrandomized controlled trial at the National Institutes of Health Clinical Center. They received a combination of carfilzomib, lenalidomide, and dexamethasone (KRd) for 8 cycles every 28 days, followed by lenalidomide maintenance for 2 years (NCT01572480) [11–13]. To be eligible for the study, patients must have had a diagnosis of SMM per the International Myeloma Working Group (IMWG) [14] and met the criteria for high-risk disease based on the 2008 Mayo Clinic, the PETHEMA group, or the Rajkumar, Landgren, and Mateos criteria [15–17]. TCR sequencing was performed on SMM samples at baseline, after cycle 1 (n = 54), cycle 4 (n = 50), cycle 8 (n = 46), cycle 20 (n = 36), and cycle 32 (n = 22) beginning November 1, 2022. The authors had access to information that could identify individual participants during data collection.

Minimal residual disease (MRD) assessment for clinical trial participants was performed on bone marrow aspirates using validated 8-color multiparametric flow cytometry consistent with the EuroFlow criteria with a sensitivity of $10^{-5}$ [18]. A population of 20 or more abnormal plasma cells defined MRD positivity with an acquisition of 3 million or greater events for a limit of detection of 0.0007%. Sustained MRD negativity was defined as the absence of MRD on two consecutive assessments at least one year apart.

For comparison, TCR sequencing data of whole blood (n = 82) or PBMCs (n = 84) were obtained from 166 healthy participants aged 23 years and older from 2 previously published studies [19,20]. Due to differences in the quantification of CDR3β molecules between versions of the TCR sequencing assay, only data generated by version 3 or 4 of the ImmunoSeq kit were included in the final analysis. Age-matching was not prospectively implemented during cohort selection, as the healthy donor data were obtained from existing datasets. However, to mitigate confounding, we excluded healthy donors under 40 years of age and adjusted for age in all multivariate analyses involving TCR diversity.

## TCR sequencing

DNA was extracted from PBMCs using the Qiagen AllPrep DNA/RNA Mini Kit (Germantown, MD, USA) following the manufacturer's instructions. To sequence the CDR3 region of the human TCR β chain (TCRB), libraries were prepared using version 3 or 4 of the ImmunoSeq hsTCRB kit (Adaptive Biotechnologies, Seattle, WA, USA) in accordance with the manufacturer's protocol. Sequencing was performed on the Illumina NextSeq 500 system using the NextSeq 500/550 Mid Output v2 kit (150 cycles, San Diego, CA, USA), following the manufacturer's guidelines. Processed sequencing files are available through ImmuneACCESS data portal: https://clients.adaptivebiotech.com/pub/coffey-2025-s (DOI 10.21417/DGC2025S).

## Statistical analysis

TCRB V, D, and J gene alignment and calculation of complementarity-determining regions 3 (CDR3) frequency was performed using the ImmunoSEQ Analyzer (Adaptive Biotechnology). Only in-frame CDR3β chain amino acids without an early stop codon were considered for subsequent analyses. To normalize the wide variability in sequencing depth, repertoires were downsampled using the immunarch (v1.0.0) R package to the number of clonotypes in the smallest repertoire without any probabilistic simulation [21]. The diversity of functional TCRB amino acid sequences was assessed using the Gini coefficient [22]. A Gini coefficient value approaching 1 indicates lower diversity, with a few dominant TCR clonotypes, signifying higher clonality (S2 Table). Wilcoxon Rank Sum test was performed to compare the Gini coefficients across patient groups with two-sided $\alpha = 0.05$. A multivariate linear regression analysis was performed to assess the relationship between Gini coefficient and age and diagnosis, using the lm function in R (v4.2.3). Public TCRB sequences were queried in the VDJdb [23], McPAS-TCR [24], PIRD TBAdb [25], and LymphoSeqDB [26] databases. Amino acid physical properties were computed using the alakazam (v1.3.0) R package [27].

TCRB clustering based on similar CDR3β amino acid sequences was performed using ClusTCR [28]. Group-specific TCRB cluster exclusivity was evaluated by permutation testing, in which baseline sample labels were shuffled 1,000 times while preserving group sizes, and observed exclusivity was compared against the resulting null distribution to derive empirical p-values. To evaluate whether distinct TCRB clusters could distinguish plasma cell dyscrasias from healthy controls, we first divided the full dataset into two groups: 70% of the samples were assigned to a training set for feature selection and model development, while the remaining 30% were held out as an independent test set for external validation. The train/test split was stratified to preserve the class distribution of healthy and diseased samples and was fixed across all analyses. Within the training dataset, we performed a differential abundance analysis using two-sided Fisher Exact tests to identify clusters with significantly different prevalence between healthy individuals and patients with plasma cell dyscrasias. Each test was conducted independently for a given cluster, and a nominal significance threshold of $\alpha = 0.001$ was used to identify candidate clusters for downstream modeling. This step was restricted to the training data only to avoid information leakage into the test set.

We then assessed the predictive utility of the selected TCRB clusters using six machine learning algorithms: elastic net regularization using glmnet (v4.1-8), random forest (randomForest, v4.7-1.2), support vector machine with radial basis function kernel (kernlab, v0.9-33, via svmRadial in caret), neural network (nnet, v7.3-19), gradient boosting machine (gbm, v2.2.2), and k-nearest neighbors (class, v7.3-22) (code available in Supplemental Methods). A two-stage evaluation framework was implemented to measure model performance. First, we conducted internal validation using a repeated stratified hold-out procedure with five iterations of random 80/20 train/test splits within the training set. For each model and iteration, we calculated the area under the receiver operating characteristic curve (AUROC), accuracy, sensitivity, specificity, and Cohen's Kappa statistic. Performance metrics were summarized across repeats using the mean and empirical 95% confidence intervals. Pairwise DeLong tests were used to compare AUROC values between models. To assess generalizability, each model was retrained on the full training dataset and evaluated on the held-out 30% test set, which was not used during feature selection or training. Variable importance scores were computed using the varImp function in the caret package (v6.0-94).

## Results

### No significant difference in TCRB repertoire diversity differences across diagnoses or treatment response

Bulk TCRB sequencing of peripheral blood was analyzed from 447 samples collected from 80 patients with MGUS, 55 with SMM, and 31 with MM and compared to those from 166 healthy donors obtained from the Adaptive ImmuneAccess database (Fig 1). The median total number of productive CDR3β amino acid sequences per sample was 204,148 (range 10,833–815,687 per sample). To account for the wide variability in the number of productive sequences, likely reflecting technical variations in sample processing and sequencing depth, each sample was downsampled to 10,833 sequences (the minimum observed) for subsequent diversity metric calculations (S1 Fig in S1 File). Furthermore, we observed a significant, moderate correlation between age and TCRB clonality measured by the Gini coefficient (S2 Fig in S1 File). To account for this confounding effect, we excluded samples from healthy donors under 40 years of age, leaving 44 control samples for the subsequent repertoire analysis. Within this cohort, the median age of healthy controls was 55 years, compared to 58 years for MGUS, 60 years for SMM, and 60 years for MM.

Although univariate analysis (Wilcoxon test) revealed a significant difference in T cell diversity (Gini coefficient) between healthy individuals and those with MGUS or SMM, this difference was not significant after adjusting for age in the multivariate linear model (Fig 2A). Additionally, we did not observe a significant difference in T cell diversity based on ImmunoSeq kit version (S3 Fig in S1 File). We further confirmed the stability of TCRB diversity in MGUS patients across a median time span of 677 days (range: 180–1149 days), demonstrating that the repertoire remained consistent over time (Fig 2B). There was no significant difference in changes within the first 600 days compared to later time points (S4 Fig in S1 File). Additionally, we found no significant association between TCRB repertoire diversity and the depth of response to induction therapy in NDMM patients (Fig 2C). Created in BioRender. Coffey, D. (2025) https://BioRender.com/i79b956

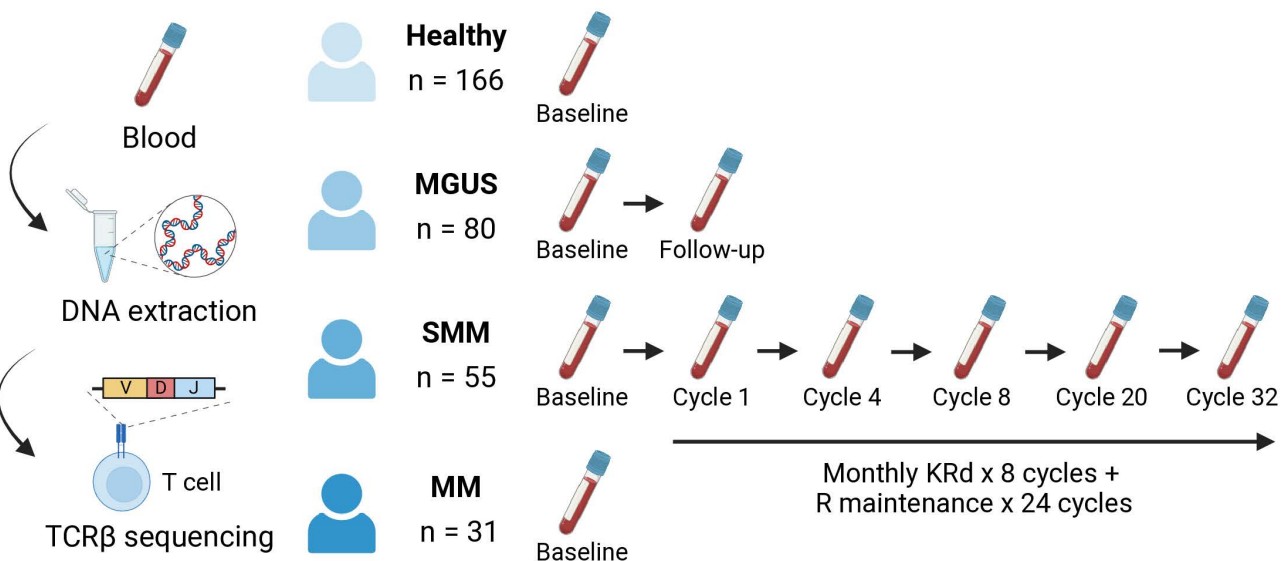

**Fig 1. Overview of experimental methods.** Bulk T-cell receptor β sequencing of DNA extracted from peripheral blood mononuclear cells (PBMCs) was analyzed from 447 samples, including 80 individuals with monoclonal gammopathy of undetermined significance (MGUS), 55 with smoldering multiple myeloma (SMM), and 31 with newly diagnosed multiple myeloma (NDMM) and compared to data from 166 healthy donors obtained from the Adaptive ImmuneAccess database. In total, 612 samples were analyzed. This included follow-up samples from 72 MGUS patients (median follow-up 677 days) and longitudinal samples from high-risk SMM patients at the end of cycle 1 (n = 54), cycle 4 (n = 50), cycle 8 (n = 46), cycle 20 (n = 36), and cycle 32 (n = 22) of carfilzomib, lenalidomide, and dexamethasone (KRd).

To investigate the impact of therapeutic intervention on TCRB repertoire, we evaluated the TCRB diversity in patients with high-risk SMM before treatment (baseline) and then again after cycles 1, 4, 8, 20, and 32 of KRd. Among the pairwise comparisons for each timepoint, no significant changes in repertoire diversity were observed compared to baseline. However, there was a trend toward reduced diversity following cycle 1 compared to baseline, and significant changes in repertoire diversity were found between cycle 1 and cycles 8 and 32, even after correcting for age (Fig 2D). However, we did not detect a difference across patients achieving sustained MRD negativity (Fig 2E). Additionally, at the individual level, we did not observe a strong trend in increasing or decreasing diversity following each cycle of treatment (Fig 2F).

### Machine learning algorithm identifies distinct TCRB clusters in patients with plasma cell dyscrasias compared to healthy donors

Previous studies have demonstrated that different TCRs recognizing the same antigen often share similar CDR3β amino acid sequences [29,30]. Based on these observations, various computational algorithms have been developed to cluster CDR3β sequences, with each cluster potentially recognizing a common antigen [28,30–33]. We hypothesized that distinct TCR clusters may be associated with plasma cell dyscrasias and could be distinguished from those in healthy blood.

To test this hypothesis, we used the ClusTCR algorithm to group TCRB sequences into clusters based on shared amino acid sequence motifs [28]. This method was chosen over other algorithms due to its superior computational efficiency on large datasets. Using only baseline samples from 166 healthy donors and 166 patients with plasma cell dyscrasias, a total of 45,571,363 CDR3β sequences were grouped into 1,240,017 clusters (median of 57,822 clusters per sample; range 154−159,529 clusters per sample). The mean cluster size was 24 CDR3β sequences (range 2−1,909 CDR3β sequences per cluster). The computed Gini coefficient of TCRB clusters was positively correlated with the Gini coefficient of TCRB sequences (Spearman correlation 0.762, S5 Fig in S1 File).

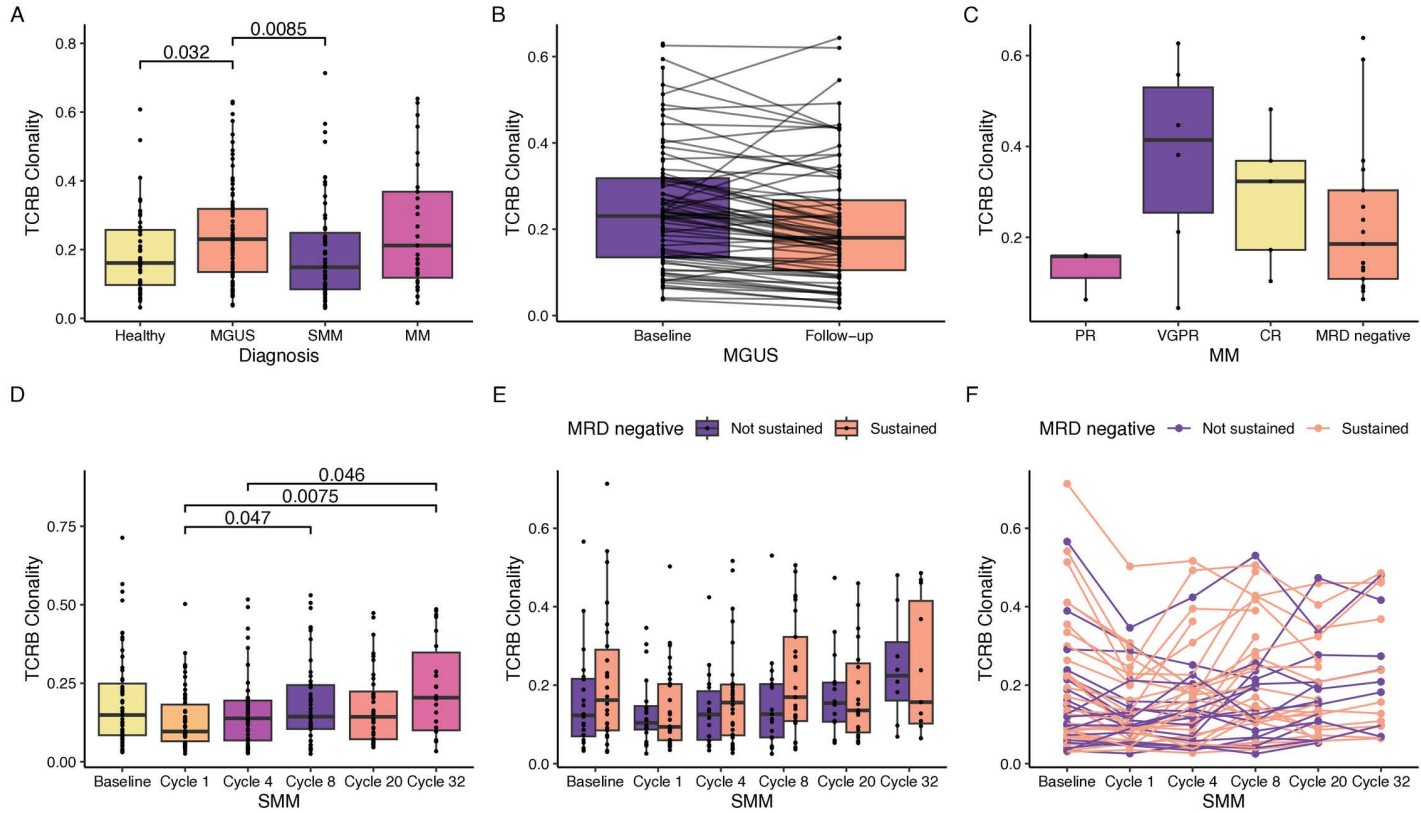

**Fig 2. T-cell receptor β chain (TCRB) clonality is not significantly associated with disease burden or depth of treatment response.** A) Comparison of TCRB clonality (Gini coefficient) between 44 healthy individuals (age > 40 years) and patients with monoclonal gammopathy of undetermined significance (MGUS), smoldering multiple myeloma (SMM), and multiple myeloma (MM). P-values of significant differences in the univariate analysis (Wilcoxon test) are shown. No significant differences were observed in the multivariate linear model that corrected for age. B) TCRB clonality in MGUS patients at baseline and follow-up (median follow-up 677 days). C) TCRB clonality in MM patients before treatment, stratified by post-treatment depth of response. D) TCRB clonality in SMM patients undergoing carfilzomib, lenalidomide, and dexamethasone (KRd) shown for the entire cohort, and E) stratified by depth of response. F) Line plot comparing the change in individual TCRB clonality among SMM patients undergoing KRd treatment. MRD, minimal residual disease; CR, complete response; VGPR, very good partial response; PR, partial response.

Next, we developed a machine learning algorithm to predict whether a sample originated from a patient with plasma cell dyscrasia or a healthy individual. To achieve this, we divided the samples into two groups: 70% for training the algorithm and 30% for testing its performance. In each group, we maintained a balanced proportion of samples from healthy individuals and patients with plasma cell dyscrasia to avoid bias and ensure the model learned patterns from both classes effectively. There was no significant difference in the number of clusters between healthy individuals (55,197) and those with plasma cell dyscrasias (58,965) in the testing dataset (Wilcoxon rank-sum test p-value = 0.137). Overlap analyses between baseline sample clusters demonstrated that 21.3% are exclusive to healthy, 10.7% to MGUS, 8.6% to SMM, and 3.4% to MM (Fig 3A). A permutation test confirmed that the observed exclusivity percentages were consistent with chance expectations for MGUS (10.7% observed vs. 10.0% expected, p = 0.42) and MM (3.4% vs. 3.6%, p = 0.77) while SMM (8.6% vs. 6.6%, p = 0.016) and healthy (21.3% vs. 24.4%, p = 0.04) showed evidence of enrichment beyond random expectations. We applied the Fisher Exact test to identify 507 differentially abundant TCRB clusters that were significantly more prevalent in either the healthy or diseased baseline samples (S3 Table). Given the large number of clusters tested, no individual cluster met a false discovery rate (FDR) threshold of < 0.05. Therefore, we used a nominal p-value threshold

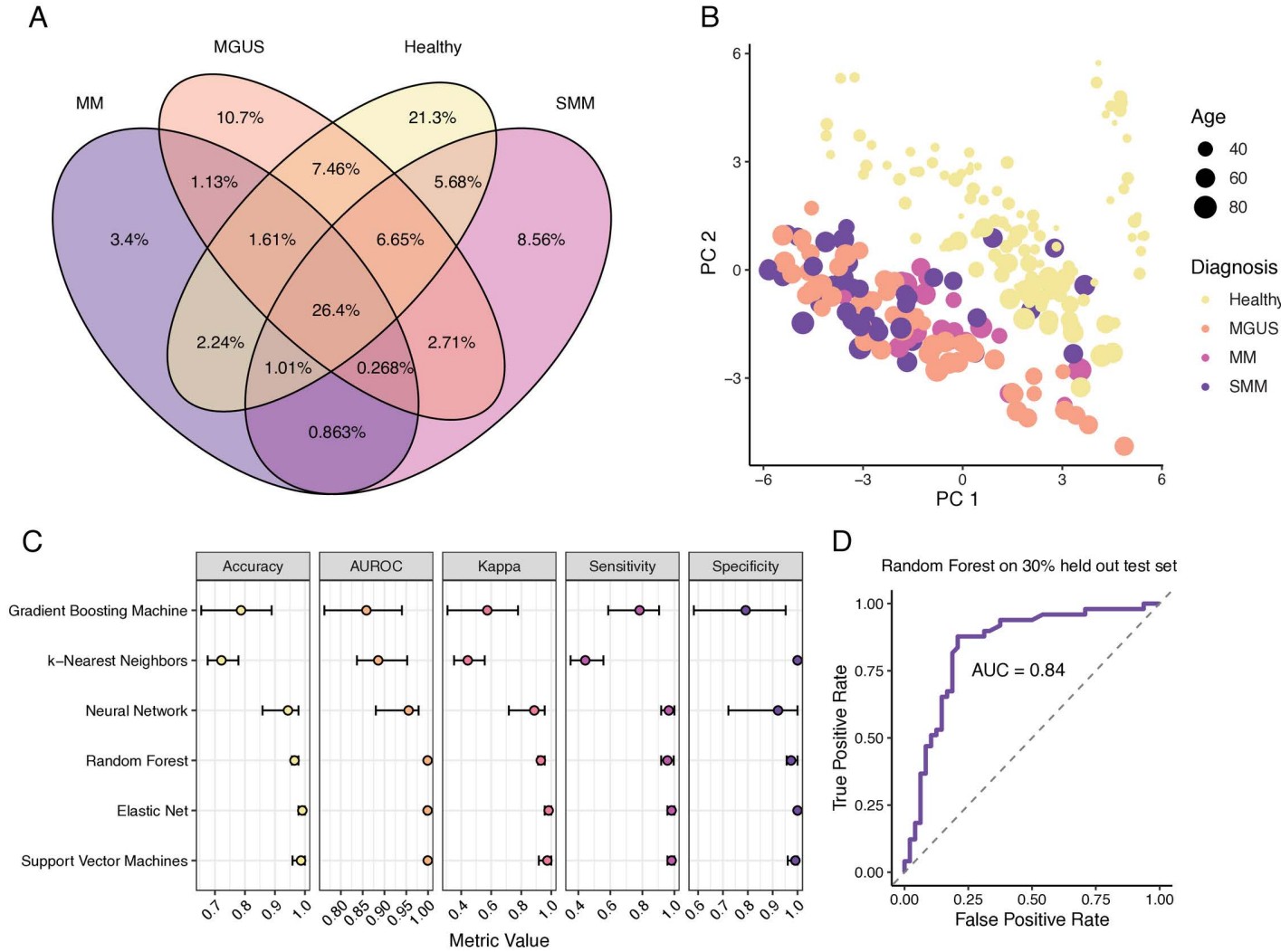

**Fig 3. TCRB clustering analysis differentiates plasma cell dyscrasias from healthy blood samples.** A) Venn diagram illustrating the overlap of all TCRB clusters between healthy, monoclonal gammopathy of undetermined significance (MGUS), smoldering multiple myeloma (SMM), and multiple myeloma (MM) baseline samples. B) Principal Component Analysis (PCA) plot visualizing the variance in frequency of differentially abundant TCRB clusters across sample types. C) Performance metrics and 95% confidence intervals for each classifier across five repeated stratified 80/20 train/test splits within the 70% training dataset. For each machine learning model, performance was evaluated using five independent train/test iterations. D) ROC curve showing the discriminative performance of the final random forest classifier when applied to the independent 30% test set that was not used during model training or feature selection. The model achieved an AUROC (area under the receiver operating characteristic curve) of 0.84, demonstrating its ability to generalize to unseen samples.

of < 0.001 for exploratory identification of candidate clusters. Notably, many of the clusters identified by p-value also exhibited moderate to high variable importance scores in the neural network model, supporting their collective relevance to disease classification, despite the absence of strong univariate effects (S3 Table).

Although the healthy donors were younger on average than the patients with plasma cell dyscrasias, the age distribution within each of the differentially abundant TCRB clusters did not significantly deviate from the overall age distribution of healthy donors in the training dataset (S6 Fig in S1 File). Principal component analysis showed good separation between healthy individuals compared to those with plasma cell dyscrasias regardless of age (Fig 3B). To validate the discriminative

ability of TCR sequence-based features, we employed a two-stage machine learning framework incorporating both internal and external validation. First, we trained six machine learning classifiers on 70% of the samples using repeated stratified hold-out resampling (5 x 80/20 splits) to estimate performance variability. For each model, we calculated the mean AUROC, accuracy, sensitivity, specificity, and Cohen's Kappa, along with 95% confidence intervals. The random forest, elastic net, and support vector machine models consistently achieved the highest average performance (mean AUROCs ≥ 0.98), while the gradient boosting machine, neural network, and k-nearest neighbors classifiers performed less well (Fig 3C). Pairwise DeLong tests indicated no statistically significant AUROC differences between the top three models, though elastic net outperformed the neural network and gradient boosting machine in several resampling iterations (S4 Table).

To evaluate generalizability, we retrained each model on the full 70% training set and assessed performance on a held-out 30% test set that had not been used during model development or feature selection. The random forest classifier achieved the best external performance, with an AUROC of 0.859, accuracy of 0.814, sensitivity of 0.796, specificity of 0.833, and Kappa of 0.629 (Fig 3D). These results confirm that disease-associated TCRB clusters encode sufficient discriminatory signal to distinguish patients with plasma cell dyscrasias from healthy individuals, even when tested on unseen data.

Finally, we computed the average physical properties of the amino acid sequences comprising each differentially abundant TCRB cluster (Fig 4A). This revealed significant differences in nearly all physical properties between TCR clusters from healthy individuals and those with plasma cell dyscrasias, suggesting structural variations in the TCRB repertoire that distinguish these groups (Fig 4B). To further investigate these 507 differentially detected clusters, we queried them against the VDJdb [23], McPAS-B [24], PIRD TBAdb [25], and LymphoSeqDB [26] databases to identify any known antigen specificities. This search uncovered 111 CDR3β amino acid sequences previously reported to be associated with microbial (e.g., cytomegalovirus, Epstein-Barr virus, influenza, tuberculosis, SARS-CoV-2, yellow fever), autoimmune (e.g., multiple sclerosis, type 1 diabetes, celiac disease, rheumatoid arthritis), or cancer-related (e.g., melanoma, neoantigen) antigens. Fig 4C presents a circle packing plot that illustrates the hierarchical relationships among these clusters. In this visualization, outer grey circles represent individual TCRB clusters, and inner colored circles denote CDR3β sequences with known antigen specificity, colored by disease or antigen type. Notably, several TCRB clusters contained distinct CDR3β sequences that shared the same antigen specificity, such as cytomegalovirus, underscoring the method's ability to group functionally convergent TCRs based on structural similarity. Colored circles not enclosed within grey boundaries indicate singleton clonotypes with known specificity that did not cluster with others. These annotated sequences were observed across both healthy and plasma cell dyscrasia samples, suggesting shared antigen exposures or public T-cell responses rather than disease-specific enrichment.

## Discussion

In this study, we leveraged high-throughput sequencing of the TCR CDR3β region to evaluate T cell repertoire diversity in patients with plasma cell dyscrasias. While earlier studies using lower-resolution techniques suggested a link between clonal T-cell expansions and improved survival [8], our findings, in line with some recent research [10], challenge this association. Our analysis did not identify significant differences in TCRB diversity between healthy participants and individuals diagnosed with MGUS, SMM, or MM after adjusting for age. Additionally, we observed stability of the T cell repertoire in MGUS patients over a median follow-up of 677 days. Furthermore, neither the depth of response to induction therapy in newly diagnosed MM patients nor the achievement of sustained MRD negativity in high-risk SMM patients correlated with TCRB diversity.

Although we noted some changes in TCRB repertoire diversity in high-risk SMM patients during KRd treatment that met statistical significance, overall, the TCRB repertoire diversity remained stable compared to baseline, and these changes were not associated with clinical outcomes. This suggests that while treatment may influence the T-cell repertoire, this measure does not necessarily translate into a clinically meaningful impact on disease progression or response. Further

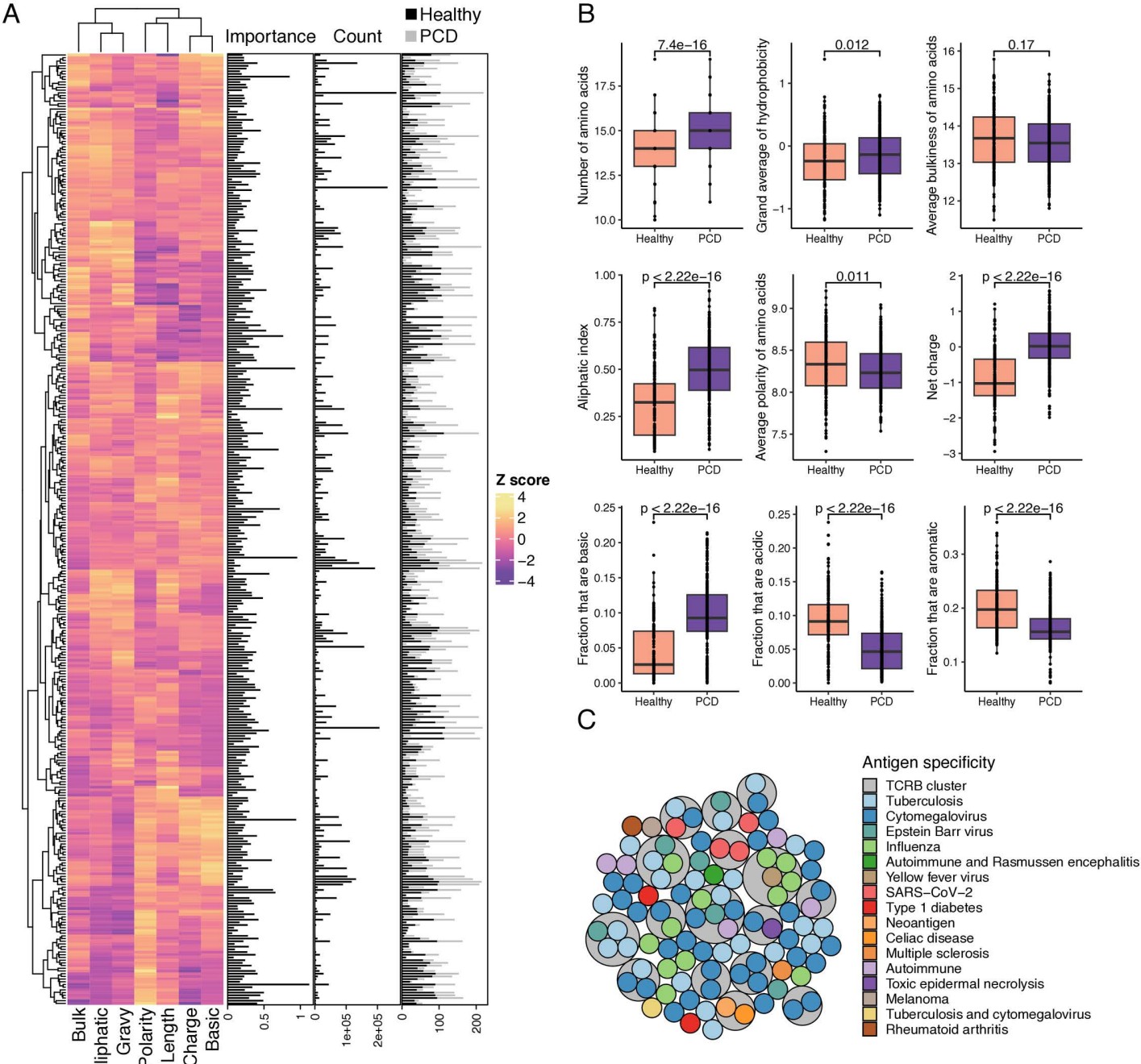

**Fig 4. Physical properties of differentially abundant TCRB clusters vary across healthy and plasma cell dyscrasia (PCD) samples.** A) Heatmap illustrating the normalized mean physical properties (columns) of CDR3β amino acid sequences within each of the differentially abundant TCRB clusters (rows). Bar plots adjacent to the heatmap display the variable importance in the neural network model, the count of CDR3β amino acid sequences per cluster, and the number of patients per diagnostic group. B) Box plots comparing the distributions of physical properties between sample types, with Wilcoxon rank-sum test p-values indicated. C) Circle packing plot illustrating the hierarchical relationship of TCRB clusters containing one or more CDR3β amino acid sequences with known antigen or pathologic condition specificity previously reported in the literature. Outer grey circles represent TCRB clusters, while inner circles denote CDR3β amino acid sequences colored by antigen specificity. Colored circles not enclosed by grey circles signify the sole clonotype within that cluster with established antigen specificity.

studies with larger cohorts and longer follow-up periods are needed to better understand the clinical relevance of these observations.

The absence of a correlation between global TCRB diversity and disease stage or treatment outcomes implies that overall diversity may be an insufficient marker of effective immune surveillance in plasma cell dyscrasias. These findings suggest that failure of immune control may be driven more by qualitative features of the T-cell response, such as antigen specificity, clonal expansion of disease-relevant T cells, or functional exhaustion, rather than by the total number or evenness of distinct TCRs. It is also possible that disease-associated immune changes are more pronounced within specific tissue compartments or T-cell subsets that are not captured by bulk peripheral repertoire metrics.

Although overall TCRB diversity was not associated with clinical outcomes, our machine learning model trained on differentially abundant clusters was able to distinguish healthy individuals from patients with plasma cell dyscrasias, underscoring that qualitative rather than quantitative TCR features are driving these differences. These differentially abundant clusters exhibited different amino acid properties, suggesting potential differences in antigen recognition albeit none were identified in this study. The biological significance of these differentially abundant clusters remains uncertain, but several possibilities may explain their enrichment in plasma cell dyscrasias. First, these clusters could represent tumor-reactive T cells that have undergone clonal expansion in response to malignant or pre-malignant plasma cell antigens. Second, they may reflect bystander T-cell expansions driven by chronic immune activation, microbial exposure, or tissue remodeling. Third, differences in the TCRB repertoire between patients and controls could be influenced by non-disease factors such as subclinical infections or vaccination history. However, we acknowledge that information regarding recent infections, inflammatory status, and prior immunization was not systematically collected for all study patients. Additionally, they could represent public or convergent TCRs that are disproportionately represented in the disease setting due to antigen-independent selection pressures. Notably, most of the differentially abundant clusters we identified did not map to known antigens. This likely reflects the significant gaps in current TCR-antigen reference databases rather than a true lack of disease relevance. These unannotated, disease-associated clusters may in fact represent T-cell responses to novel tumor neoantigens or other undiscovered disease-specific epitopes, making them high-priority candidates for future functional investigation. Nevertheless, several clusters contained distinct CDR3β sequences with known specificity for the same antigen, suggesting potential convergent immune responses.

In addition to prospective studies to validate the accuracy of this machine learning methodology for discriminating between healthy individuals and those with plasma cell dyscrasias, future studies should focus on identifying the antigenic targets of the TCRs that comprise the differentially abundant TCRB clusters identified in this study. This could provide valuable insights into the mechanisms underlying the immune response in plasma cell dyscrasias and potentially reveal novel therapeutic targets. Additionally, exploring the phenotypic states and functional properties of these distinct T-cell populations could further elucidate their role in disease pathogenesis and response to therapy.

In conclusion, although this study did not demonstrate a significant correlation between total T-cell repertoire diversity and treatment outcomes in plasma cell dyscrasias, we successfully identified unique TCRB clusters that distinguish healthy individuals from patients with plasma cell disorders. To our knowledge, this represents the largest analysis of the TCRB repertoire in plasma cell dyscrasias to date and the first to show that specific TCRB clusters can distinguish patients with these disorders from healthy controls. Additional studies are warranted to clarify the functional significance of these clusters and their potential involvement in disease initiation and progression. These findings underscore the importance of ongoing research into the relationship between T-cell responses and clinical outcomes.

## Supporting information

**S1 File. S1 Fig. TCR repertoire downsampling normalizes variation and maintains clonality distribution.** A) Box plot comparing the total number of productive rearrangements per sample, categorized by patient diagnosis (n = 612 samples). Wide variability in productive rearrangements is observed across samples and diagnoses, likely reflecting technical

variations in sample processing and sequencing depth. Wilcoxon rank-sum test p-values are indicated as follows: *** p<0.001, ** p<0.01, * p<0.05. B) Downsampling of TCRB repertoires reduces the variation in clonality (measured by Gini coefficient) while maintaining the overall distribution across sample types. This normalization step ensures unbiased comparison of TCRB clonality between different groups. Monoclonal gammopathy of undetermined significance (MGUS), smoldering multiple myeloma (SMM), and multiple myeloma (MM). **S2 Fig. TCRB repertoire diversity decreases with age.** A) Scatter plot comparing age to TCRB clonality (measured by Gini coefficient) in healthy individuals (n = 166) to baseline samples from patients with monoclonal gammopathy of undetermined significance (MGUS, n = 80), smoldering multiple myeloma (SMM, n = 55), and multiple myeloma (MM, n = 31). A moderate positive correlation exists between age and TCRB clonality, particularly evident in healthy and MGUS groups due to their wider age range. B) Subgroup analysis of TCRB repertoire diversity by age and disease state. Healthy individuals aged 40–49 years exhibit significantly greater TCRB diversity compared to MGUS patients of similar age. Within the 60–69 age group, MGUS patients display significantly higher TCRB clonality (lower diversity) than SMM patients. P-values were calculated using the Wilcoxon rank-sum test. *** = 0.001, ** = 0.01, * = 0.05, NS = not significant. **S3 Fig. The version of the ImmunoSeq kit does not impact TCRB repertoire diversity.** Box plots display the distributions of age, number of productive rearrangements, original TCRB clonality, and downsampled TCRB clonality across individuals sequenced with ImmunoSeq version 3 (V3, n = 84) and version 4 (V4, n = 82) among healthy donors. P-values were calculated using the Wilcoxon rank-sum test. *** = 0.001, ** = 0.01, * = 0.05, NS = not significant. **S4 Fig. Longitudinal stability of the T cell receptor beta (TCRB) repertoire in patients with monoclonal gammopathy of undetermined significance (MGUS).** A) Percent change in TCRB clonality between two time points for 72 MGUS patients with at least two samples collected over time. Each line represents an individual patient. While a trend toward increased TCRB diversity over time is observed for the majority of patients, B) a paired Wilcoxon rank-sign test reveals no statistically significant difference in clonality changes within the first 600 days compared to later time points. NS = not significant. **S5 Fig. Correlation between TCRB CDR3β amino acid sequence diversity and TCRB cluster diversity.** The Gini coefficient, a measure of inequality, was calculated for the frequency of TCRB CDR3β amino acid sequences and the frequency of TCRB clusters (identified using the ClusTCR method). A higher Gini coefficient indicates greater diversity in the TCR repertoire. Spearman correlation 0.762. **S6 Fig. Age distribution of healthy individuals across all differentially abundant clusters used in the training dataset.** To assess potential age bias in the TCRB cluster composition, we compared the age distribution of healthy individuals within each differentially abundant cluster to the age distribution of all healthy individuals (n = 166) in the training dataset. Using a Wilcoxon Rank Sum test with FDR correction, we found no significant differences in age distributions between any cluster and the overall healthy population. This indicates that the identified TCRB clusters are not driven by age-specific variations.
(PDF)

**S1 Table. Individual sample characteristics (n = 612).** Minimal residual disease (MRD) assessment is shown for smolder multiple myeloma (SMM) samples after cycles 1, 4, 8, 20, and 32 of carfilzomib, lenalidomide, and dexamethasone (KRd). Deepest response assessed by the International Myeloma Working Group following induction therapy for multiple myeloma (MM) is shown. Monoclonal gammopathy of undetermined signifiance (MGUS) samples are from diagnose and follow-up of untreated patients. Healthy samples were obtained from two pubicaly available datasets. sCR, stringent complete response; CR, complete response; VGPR, very good partial response; PR, partial response.
(PDF)

**S2 Table. Number of unique productive sequences, Gini coefficient, and downsampled Gini coefficient computed per sample.** Only healthy donors ≥ 40 years of age were used in the final analysis.
(PDF)

**S3 Table. Differentially abundant TCRB clusters. Mean physical property for each cluster are shown.** Sample count are provided for healthy and plasma cell dyscrasia (PCD) patients. Wilcoxon Rank Sum test P value and false

discovery rate (FDR) are reported for each cluster along with top performing machine learning model variable importance rank (1 is the most important and 507 is the least). Length, number of amino acids; gravy, grand average of hydrophobicity index; bulk, average bulkiness of amino acids; aliphatic, aliphatic index; polarity, average polarity of amino acids; charge, net charge; basic, fraction of informative positions that are Arg, His or Lys; acidic, fraction of informative positions that are Asp or Glu; aromatic, fraction of informative positions that are His, Phe, Trp or Tyr.
(PDF)

**S4 Table. Performance metrics for each classifier across five repeated stratified 80/20 train/test splits.** For each machine learning model, performance was evaluated using five independent train/test iterations. AUROC area under the receiver operating characteristic curve; rf, Random Forest; svmRadial, Support Vector Machine with Radial Basis Function Kernel; glmnet, Elastic Net Regularization; nnet, Neural Network; gbm, Gradient Boosting Machine; knn, k-Nearest Neighbors.
(PDF)

## Author contributions

**Conceptualization:** Andrea C. Baines, Dickran Kazandjian.

**Data curation:** Yong Zhang, Elizabeth Hill.

**Formal analysis:** David Coffey, Yong Zhang.

**Methodology:** David Coffey, Yong Zhang.

**Project administration:** Frank Cross Jr.

**Supervision:** Reena Philip.

**Visualization:** David Coffey, Yong Zhang.

**Writing – original draft:** David Coffey, Yong Zhang.

**Writing – review & editing:** David Coffey, Yong Zhang, Elizabeth Hill, Reena Philip, Marc R. Theoret, Ola Landgren, Andrea C. Baines, Dickran Kazandjian.

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
