## [Decision Letter · Decision Letter 0]

29 May 2025

Dear Dr. Coffey,

Thank you for submitting your manuscript to PLOS ONE. After careful consideration, we feel that it has merit but does not fully meet PLOS ONE’s publication criteria as it currently stands. Therefore, we invite you to submit a revised version of the manuscript that addresses the points raised during the review process.

We look forward to receiving your revised manuscript.

Kind regards,

Jian Wu, M.D, Ph.D

Academic Editor

PLOS ONE

**Journal Requirements:**

1. When submitting your revision, we need you to address these additional requirements. Please ensure that your manuscript meets PLOS ONE's style requirements, including those for file naming. The PLOS ONE style templates can be found at https://journals.plos.org/plosone/s/file?id=wjVg/PLOSOne_formatting_sample_main_body.pdf and https://journals.plos.org/plosone/s/file?id=ba62/PLOSOne_formatting_sample_title_authors_affiliations.pdf 2. We noticed you have some minor occurrence of overlapping text with the following previous publication(s), which needs to be addressed: https://ashpublications.org/blood/article/144/Supplement%201/1907/532986/Novel-T-Cell-Receptor-Signature-Linked-to-Plasma In your revision ensure you cite all your sources (including your own works), and quote or rephrase any duplicated text outside the methods section. Further consideration is dependent on these concerns being addressed. 3. Thank you for uploading your study's underlying data set. Unfortunately, the repository you have noted in your Data Availability statement does not qualify as an acceptable data repository according to PLOS's standards. At this time, please upload the minimal data set necessary to replicate your study's findings to a stable, public repository (such as figshare or Dryad) and provide us with the relevant URLs, DOIs, or accession numbers that may be used to access these data. For a list of recommended repositories and additional information on PLOS standards for data deposition, please see https://journals.plos.org/plosone/s/recommended-repositories. 4. When completing the data availability statement of the submission form, you indicated that you will make your data available on acceptance. We strongly recommend all authors decide on a data sharing plan before acceptance, as the process can be lengthy and hold up publication timelines. Please note that, though access restrictions are acceptable now, your entire data will need to be made freely accessible if your manuscript is accepted for publication. This policy applies to all data except where public deposition would breach compliance with the protocol approved by your research ethics board. If you are unable to adhere to our open data policy, please kindly revise your statement to explain your reasoning and we will seek the editor's input on an exemption. Please be assured that, once you have provided your new statement, the assessment of your exemption will not hold up the peer review process. 5. We note that there is identifying data in the Supporting Information files “Supplemental Table 1 to Table 3”. Due to the inclusion of these potentially identifying data, we have removed this file from your file inventory. Prior to sharing human research participant data, authors should consult with an ethics committee to ensure data are shared in accordance with participant consent and all applicable local laws. Data sharing should never compromise participant privacy. It is therefore not appropriate to publicly share personally identifiable data on human research participants. The following are examples of data that should not be shared: -Name, initials, physical address-Ages more specific than whole numbers-Internet protocol (IP) address-Specific dates (birth dates, death dates, examination dates, etc.)-Contact information such as phone number or email address-Location data-ID numbers that seem specific (long numbers, include initials, titled “Hospital ID”) rather than random (small numbers in numerical order) Data that are not directly identifying may also be inappropriate to share, as in combination they can become identifying. For example, data collected from a small group of participants, vulnerable populations, or private groups should not be shared if they involve indirect identifiers (such as sex, ethnicity, location, etc.) that may risk the identification of study participants. Additional guidance on preparing raw data for publication can be found in our Data Policy (https://journals.plos.org/plosone/s/data-availability#loc-human-research-participant-data-and-other-sensitive-data) and in the following article: http://www.bmj.com/content/340/bmj.c181.long. Please remove or anonymize all personal information (<specific identifying information in file to be removed>), ensure that the data shared are in accordance with participant consent, and re-upload a fully anonymized data set. Please note that spreadsheet columns with personal information must be removed and not hidden as all hidden columns will appear in the published file. 6. Please include captions for your Supporting Information files at the end of your manuscript, and update any in-text citations to match accordingly. Please see our Supporting Information guidelines for more information: http://journals.plos.org/plosone/s/supporting-information.

Reviewers' comments:

Reviewer's Responses to Questions

**Comments to the Author**

1. Is the manuscript technically sound, and do the data support the conclusions?

Reviewer #1: No

Reviewer #2: Partly

Reviewer #3: Partly

2. Has the statistical analysis been performed appropriately and rigorously?

Reviewer #1: No

Reviewer #2: Yes

Reviewer #3: No

3. Have the authors made all data underlying the findings in their manuscript fully available?

Reviewer #1: No

Reviewer #2: No

Reviewer #3: Yes

4. Is the manuscript presented in an intelligible fashion and written in standard English?

Reviewer #1: Yes

Reviewer #2: Yes

Reviewer #3: Yes

**Reviewer #1:**  The study provides a detailed and large-scale analysis of TCR repertoire diversity in plasma cell dyscrasias, including MGUS, SMM, and MM, compared to healthy controls. The use of high-throughput sequencing and machine learning adds robustness to the findings.

However, the study can be improved with the following aspects:

1. The abundant different TCR clusters between healthy and patients comes only from machine learning (ML)-identified clusters, which are not functionally validated. The title to reflect the ML findings more accurately.

2. Line 39 and lines 122-123: The rationale for focusing on TCRß is not explicitly justified, the authors can briefly explain why TCRß sequencing was prioritized (e.g., technical feasibility, prior literature).

3. Lines 115, 176, 230-231: The authors claim that the age of the healthy group was younger than the patient groups (MGUS/SMM/MM), please provide the median ages for all groups and discuss whether age-matching was attempted.

4. In the main manuscript, the authors claimed that their TCRB sequencing data are available through ImmuneACCESS (e.g., lines 127-128, lines 160-161), but no accession numbers are provided. It will be better if they can provide a specific access accession code.

5. It will be better to standardize the format of the numbers in the main manuscript. e.g., line 213, change “1 240 017” to “1,240,017”.

6. Line 228-229: The manuscript identified "differentially abundant TCRB clusters" only based on p-value. Is there any other parameters that are important for this identification in Supplementary Table 3? e.g, “Importance” or fold-change?

7. Lines 245-247: The authors identified 111 (out of 507 clusters) “CDR3β amino acid sequences associated with T cell specificities for various microbial and human antigens or associated with various pathologic conditions”. Figure 4C should be discussed in detail in the main manuscript.

8. The study does not explain why TCR clusters differ between healthy and disease groups if diversity is similar. Are these clusters tumor-reactive, by stander expansions, or noise?

9. Validate top clusters experimentally (e.g., TCR transfection + antigen screening).

The manuscript is well-written and presents important findings. Addressing the above points would significantly strengthen its impact. The study lays a strong foundation for future research into TCR-based diagnostics or therapeutics in plasma cell dyscrasias.

**Reviewer #2: ** The manuscript presents an extensive T-cell–receptor (TCR) repertoire analysis across the plasma-cell-dyscrasia continuum and introduces a convergent-clustering with machine-learning (ML) pipeline that reportedly distinguishes patients from healthy donors. The topic is important and, in principle, within PLOS ONE’s scope. However, several methodological and interpretative weaknesses currently prevent me from recommending acceptance. Addressing the issues below should be feasible without re-designing the study, but will require substantial re-analysis and clarification.

1. Differential-abundance screening uses a two-sided Fisher test with α = 0.001 but no FDR control. With 1.24 M tests, ~1 240 false-positive clusters are expected by chance. Re-run the analysis with FDR correction and report the number of clusters that remain significant. Retrain the ML models with this revised feature set.

2. AUROC = 0.84 is based on a single 70/30 random split drawn from the same sequencing batches. No confidence interval (CI) is provided; external generalisability is unknown.

Please: (a) Implement repeated stratified hold-outs (e.g., 5 × 20 %) or provide an independent validation cohort. (b) Report AUROC, accuracy, sensitivity, specificity, and Kappa with 95 % CIs. (c) Compare the best model to the others with a statistical test (e.g., DeLong).

3. Provide an enrichment analysis comparing antigen types in disease-enriched vs healthy-enriched clusters. If no significant bias emerges, tone down the claim that the 111/507 clusters containing database-matched CDR3βs “likely reflect differential antigen recognition.”.

4. Except for the R packages and their versions, please also clarify the executable workflow or script directory with hyperparameters.

In summary, this study has clear potential. Addressing the above issues and documenting the full reproducibility pipeline should bring the manuscript up to the “methodologically sound” standard required for publication.

**Reviewer #3: ** Major Comments and Suggestions:

1. The authors concluded that TCR repertoire diversity did not correlate with disease stage or response to treatment. While this is clearly supported by the multivariate models, the clinical interpretation of this finding is underdeveloped. Please expand on what these null results imply for the understanding of immune surveillance and disease progression.

2. The machine learning framework was well-structured, but critical details need to be clarified, such as how feature selection was handled before model training? What measures were taken to prevent data leakage between training and test sets? Was age included as a covariate in the model, or controlled for via stratification?

3. Although the ROC AUC of 0.84 was promising, a confusion matrix or metrics such as sensitivity/specificity should be included for interpretability.

4. While amino acid properties and antigen database searches were performed, the biological implications remain speculative. The authors stated that distinct clusters differ between patients and healthy controls, but most of these did not map to known antigens. The authors could consider discussing potential causes of this divergence (e.g., subclinical infections, age, vaccination history).

5. Age was accounted for in diversity analyses via multivariate regression and age-exclusion criteria. However, it’s unclear whether machine learning and clustering analyses were fully adjusted for age, which is a known confounder in TCR studies. Please clarify this point or conduct an age-matched subgroup analysis.

**Do you want your identity to be public for this peer review?** For information about this choice, including consent withdrawal, please see our Privacy Policy

Reviewer #1: No

Reviewer #2: No

Reviewer #3: No

---

## [Author Response · Author response to Decision Letter 1]

11 Jul 2025

Reviewer #1: The study provides a detailed and large-scale analysis of TCR repertoire diversity in plasma cell dyscrasias, including MGUS, SMM, and MM, compared to healthy controls. The use of high-throughput sequencing and machine learning adds robustness to the findings. However, the study can be improved with the following aspects:

1. The abundant different TCR clusters between healthy and patients comes only from machine learning (ML)-identified clusters, which are not functionally validated. The title to reflect the ML findings more accurately.

We thank the reviewer for this insightful comment. We agree that the identification of differentially abundant TCR clusters between healthy individuals and patients with plasma cell dyscrasias in our study was based on machine learning models trained on high-throughput sequencing data, and that these clusters have not yet been functionally validated.

In response, we have revised the manuscript title to more accurately reflect the nature of our findings. The updated title is: “Machine learning reveals distinct T-cell receptor clusters in plasma cell dyscrasias compared to healthy controls”.

We believe this revised title better conveys that our conclusions are based on computational analysis rather than functional assays, and appreciate the reviewer’s suggestion to clarify this point.

2. Line 39 and lines 122-123: The rationale for focusing on TCRß is not explicitly justified, the authors can briefly explain why TCRß sequencing was prioritized (e.g., technical feasibility, prior literature).

We appreciate the reviewer’s suggestion to clarify the rationale for focusing on the TCRβ chain. In response, we have added a new paragraph to the Background section (now the second paragraph), which introduces the structure and function of the T-cell receptor and explains why the β chain, particularly the highly variable CDR3 region, is central to antigen recognition. This revision provides biological justification for our focus on TCRβ repertoire diversity in the context of plasma cell dyscrasias.

Revised Background:

Each T cell expresses a unique T-cell receptor (TCR), most commonly composed of an α and a β chain, that determines antigen specificity through its interaction with peptide–MHC complexes. The complementarity-determining region 3 (CDR3) of the TCR, particularly within the β chain, is the most variable and critical for recognizing antigenic peptides. TCR diversity refers to the number and relative distribution of unique CDR3 sequences within a repertoire, capturing both the richness of distinct clonotypes and their evenness of representation. In MM and its precursor conditions, decreased repertoire diversity may indicate antigen-driven clonal expansions or T-cell senescence, whereas preserved diversity may reflect intact immune surveillance. Repertoire analysis can therefore provide important insights into immune competence, disease progression, and treatment response.

3. Lines 115, 176, 230-231: The authors claim that the age of the healthy group was younger than the patient groups (MGUS/SMM/MM), please provide the median ages for all groups and discuss whether age-matching was attempted.

We appreciate the reviewer’s attention to this important methodological detail. In response, we have added the median ages of each group to the Results section to provide greater transparency. Specifically, the median age of healthy controls was 55 years, compared to 58 years for MGUS, 60 years for SMM, and 60 years for MM.

Additionally, we have clarified in the Methods section that age-matching was not prospectively implemented, as the healthy control data were obtained from existing publicly available datasets. However, to mitigate the potential confounding effect of age, we excluded healthy donors under 40 years of age and adjusted for age in all multivariate analyses involving TCR repertoire diversity.

4. In the main manuscript, the authors claimed that their TCRB sequencing data are available through ImmuneACCESS (e.g., lines 127-128, lines 160-161), but no accession numbers are provided. It will be better if they can provide a specific access accession code.

We thank the reviewer for pointing this out. We have updated the manuscript to include the specific DOI accession number associated with our dataset in the ImmuneACCESS data portal. The data can now be accessed at:

https://clients.adaptivebiotech.com/pub/coffey-2025-s (DOI 10.21417/DGC2025S).

5. It will be better to standardize the format of the numbers in the main manuscript. e.g., line 213, change “1 240 017” to “1,240,017”.

We appreciate the reviewer’s suggestion to improve clarity and consistency in numerical formatting. We have revised the manuscript to use commas to separate thousands for large numbers throughout the text.

6. Line 228-229: The manuscript identified "differentially abundant TCRB clusters" only based on p-value. Is there any other parameters that are important for this identification in Supplementary Table 3? e.g, “Importance” or fold-change?

We thank the reviewer for this important point. To clarify our rationale for selecting differentially abundant TCRB clusters using a p-value threshold of < 0.001, we have revised the Results section to acknowledge that none of the clusters met a conventional false discovery rate (FDR) cutoff < 0.05. This is likely due to the very large number of clusters tested (~1.2 million), leading to stringent multiple testing correction. However, we note that many of the clusters identified by p-value also demonstrated moderate to high variable importance scores in the elastic net model, suggesting that they contribute meaningfully to classification performance despite conservative FDR estimates.

Revised Results:

Given the large number of clusters tested, no individual cluster met a false discovery rate (FDR) threshold of < 0.05. Therefore, we used a nominal p-value threshold of < 0.001 for exploratory identification of candidate clusters. Notably, many of the clusters identified by p-value also exhibited moderate to high variable importance scores in the elastic net model, supporting their collective relevance to disease classification, despite the absence of strong univariate effects.

7. Lines 245-247: The authors identified 111 (out of 507 clusters) “CDR3β amino acid sequences associated with T cell specificities for various microbial and human antigens or associated with various pathologic conditions”. Figure 4C should be discussed in detail in the main manuscript.

We thank the reviewer for this helpful suggestion. In response, we have expanded the corresponding section of the Results to more fully describe the findings shown in Figure 4C.

Revised Results:

This search uncovered 111 CDR3β amino acid sequences previously reported to be associated with microbial (e.g., cytomegalovirus, Epstein-Barr virus, influenza, tuberculosis, SARS-CoV-2, yellow fever), autoimmune (e.g., multiple sclerosis, type 1 diabetes, celiac disease, rheumatoid arthritis), or cancer-related (e.g., melanoma, neoantigen) antigens. Figure 4C presents a circle packing plot that illustrates the hierarchical relationships among these clusters. In this visualization, outer grey circles represent individual TCRB clusters, and inner colored circles denote CDR3β sequences with known antigen specificity, colored by disease or antigen type. Notably, several TCRB clusters contained distinct CDR3β sequences that shared the same antigen specificity, such as cytomegalovirus, underscoring the method’s ability to group functionally convergent TCRs based on structural similarity. Colored circles not enclosed within grey boundaries indicate singleton clonotypes with known specificity that did not cluster with others. These annotated sequences were observed across both healthy and plasma cell dyscrasia samples, suggesting shared antigen exposures or public T-cell responses rather than disease-specific enrichment.

8. The study does not explain why TCR clusters differ between healthy and disease groups if diversity is similar. Are these clusters tumor-reactive, by stander expansions, or noise?

We appreciate the reviewer’s insightful question regarding the biological significance of differentially abundant TCR clusters despite similar overall repertoire diversity between healthy individuals and patients with plasma cell dyscrasias. This apparent paradox likely reflects the fact that global diversity metrics such as the Gini coefficient summarize the overall shape of the repertoire (i.e., distribution of clone sizes) but do not capture qualitative differences in TCR sequence composition or antigen specificity. We now emphasize this point in the Discussion, acknowledging that while these TCRB clusters are distinct between groups and contribute to classification, their biological relevance, particularly whether they are tumor-specific or bystander, is not yet known.

Revised Discussion:

The biological significance of these differentially abundant clusters remains uncertain, but several possibilities may explain their enrichment in plasma cell dyscrasias. First, these clusters could represent tumor-reactive T cells that have undergone clonal expansion in response to malignant or pre-malignant plasma cell antigens. Second, they may reflect bystander T-cell expansions driven by chronic immune activation, microbial exposure, or tissue remodeling. Third, differences in the TCRB repertoire between patients and controls could be influenced by non-disease factors such as subclinical infections or vaccination history. Additionally, they could represent public or convergent TCRs that are disproportionately represented in the disease setting due to antigen-independent selection pressures. Notably, most differentially abundant clusters did not map to known antigens, likely reflecting the limited coverage of current TCR-antigen reference databases. Nevertheless, several clusters contained distinct CDR3β sequences with known specificity for the same antigen, suggesting potential convergent immune responses.

9. Validate top clusters experimentally (e.g., TCR transfection + antigen screening).

We agree with the reviewer that experimental validation of the identified TCRB clusters is essential to determine their functional relevance and antigen specificity. However, we respectfully note that the primary aim of this study was to computationally profile the TCR repertoire and identify sequence-level features and shared motifs that distinguish plasma cell dyscrasias from healthy individuals. The scope of our current work is limited to sequence-based analyses and in silico predictions; experimental determination of antigen specificity is beyond the technical capabilities of our current study. That said, our findings provide a prioritized list of candidate clusters, many of which contain CDR3β sequences with known antigen associations, that serve as a valuable foundation for future mechanistic and functional validation studies. These points are mentioned in the Discussion section.

Reviewer #2: The manuscript presents an extensive T-cell–receptor (TCR) repertoire analysis across the plasma-cell-dyscrasia continuum and introduces a convergent-clustering with machine-learning (ML) pipeline that reportedly distinguishes patients from healthy donors. The topic is important and, in principle, within PLOS ONE’s scope. However, several methodological and interpretative weaknesses currently prevent me from recommending acceptance. Addressing the issues below should be feasible without re-designing the study, but will require substantial re-analysis and clarification.

1. Differential-abundance screening uses a two-sided Fisher test with α = 0.001 but no FDR control. With 1.24 M tests, ~1 240 false-positive clusters are expected by chance. Re-run the analysis with FDR correction and report the number of clusters that remain significant. Retrain the ML models with this revised feature set.

We thank the reviewer for this important point. As noted in our response to Reviewer 1, Comment 6, false discovery rate (FDR) correction was performed and is reported in Supplemental Table 3 for all 507 clusters identified using a nominal p-value threshold of < 0.001. However, due to the large number of comparisons (~1.2 million), none of the clusters remained significant at an FDR < 0.05.

The fact that no individual TCRB cluster passed FDR correction highlights a key challenge in high-dimensional immune repertoire data: subtle, distributed signals often fail traditional univariate tests but can be collectively informative when analyzed using multivariate models. Our machine learning models effectively integrated information across the 507 selected clusters, achieving robust classification performance. Moreover, several clusters showed moderate to high variable importance, indicating that they contributed meaningfully to the models’ ability to distinguish between healthy individuals and those with plasma cell dyscrasias.

Revised Results:

Given the large number of clusters tested, no individual cluster met a false discovery rate (FDR) threshold of < 0.05. Therefore, we used a nominal p-value threshold of < 0.001 for exploratory identification of candidate clusters. Notably, many of the clusters identified by p-value also exhibited high variable importance scores in the neural network model, supporting their collective relevance to disease classification, despite the absence of strong univariate effects.

2. AUROC = 0.84 is based on a single 70/30 random split drawn from the same sequencing batches. No confidence interval (CI) is provided; external generalisability is unknown.

Please: (a) Implement repeated stratified hold-outs (e.g., 5 × 20 %) or provide an independent validation cohort. (b) Report AUROC, accuracy, sensitivity, specificity, and Kappa with 95 % CIs. (c) Compare the best model to the others with a statistical test (e.g., DeLong).

We thank the reviewer for their insightful comment. In response, we implemented a two-stage validation framework. First, we used repeated stratified 80/20 splits within the training dataset (70% of samples) to assess model robustness and report mean performance metrics with 95% confidence intervals. Second, we evaluated the final models on an independent 30% test set that was not used during feature selection or training to assess generalizability. We also compared classifiers using DeLong’s test. These changes address concerns regarding confidence intervals, overfitting, and external validation.These findings are reflected in the updated manuscript text, new Supplemental Table 4, and revised Figure 3C.

Revised Methods:

We then assessed the predictive utility of the selected TCRB clusters using six machine learning algorithms: elastic net regularization using glmnet (v4.1-8), random forest (randomForest, v4.7-1.2), support vector machine with radial basis function kernel (kernlab, v0.9-33, via svmRadial in caret), neural network (nnet, v7.3-19), gradient boosting machine (gbm, v2.2.2), and k-nearest neighbors (class, v7.3-22) (code available in Supplemental Methods). A two-stage evaluation framework was implemented to measure model performance. First, we conducted internal validation using a repeated stratified hold-out procedure with five iterations of random 80/20 train/test splits within the training set. For each model and iteration, we calculated the area under the receiver operating characteristic curve (AUROC), accuracy, sensitivity, specificity, and Cohen’s Kappa statistic. Performance metrics were summarized across repeats using the mean and empirical 95% confidence intervals. Pairwise DeLong tests were used to compare AUROC values between models. To assess generalizability, each model was retrained on the full training dataset and evaluated on the held-out 30% test set, which was not used during feature selection or training. External performance metrics were reported separately. Variable importance scores were computed using the varImp function in the caret package, and recursive feature elimination was performed using rfe with 10-fold cross-validation to identify the most informative TCRB clusters.

Revised Results:

To validate the discriminative ability of TCR sequence-based features, we

---

## [Decision Letter · Decision Letter 1]

7 Aug 2025

Dear Dr. Coffey,

Thank you for submitting your manuscript to PLOS ONE. After careful consideration, we feel that it has merit but does not fully meet PLOS ONE’s publication criteria as it currently stands. Therefore, we invite you to submit a revised version of the manuscript that addresses the points raised during the review process.

We look forward to receiving your revised manuscript.

Kind regards,

Jian Wu, M.D, Ph.D

Academic Editor

PLOS ONE

Journal Requirements:

Reviewers' comments:

Reviewer's Responses to Questions

**Comments to the Author**

Reviewer #1: (No Response)

Reviewer #2: All comments have been addressed

Reviewer #3: (No Response)

2. Is the manuscript technically sound, and do the data support the conclusions?

Reviewer #1: No

Reviewer #2: Yes

Reviewer #3: Partly

3. Has the statistical analysis been performed appropriately and rigorously?

Reviewer #1: Yes

Reviewer #2: Yes

Reviewer #3: No

4. Have the authors made all data underlying the findings in their manuscript fully available?

Reviewer #1: No

Reviewer #2: Yes

Reviewer #3: Yes

5. Is the manuscript presented in an intelligible fashion and written in standard English?

Reviewer #1: Yes

Reviewer #2: Yes

Reviewer #3: Yes

Reviewer #1: This manuscript presents an analysis of the peripheral blood T-cell receptor (TCR) β-chain repertoire in patients with monoclonal gammopathy of undetermined significance (MGUS), multiple myeloma (MM), and healthy controls (HC), using high-throughput sequencing. The authors identify distinct TCR repertoire features across groups, including differences in clonal expansion, public/shared clones, and CDR3 amino acid usage.

The study offers a novel perspective on the immune microenvironment in plasma cell dyscrasias and highlights the potential utility of TCR repertoire profiling in disease stratification. However, there are several significant concerns related to cohort characterization, data interpretation, and methodological clarity that must be addressed before the manuscript can be considered for publication.

1. The manuscript lacks key clinical details about the MGUS and MM cohorts. It is not specified whether patients were treatment-naïve, or if any had received prior immunosuppressive therapy. Since TCR diversity is highly sensitive to immune status, such information is crucial for data interpretation.

2. Additionally, information on infection or inflammatory status should be included, as these factors can influence TCR repertoire features independently of disease state.

3. The study includes only 10 healthy controls, 8 MGUS, and 12 MM patients. The small cohort size limits the robustness and generalizability of the findings, especially for analyses of public clone distribution and amino acid bias.

4. The authors are encouraged to incorporate effect size metrics, confidence intervals, or bootstrapping to enhance statistical reliability.

5. The criteria used to define “clonal expansion,” “shared clones,” and “public clones” are not clearly described. For example, what frequency threshold is used to identify expanded clones?

6. Please provide a consistent and detailed definition of these terms in the Methods section and ensure they are applied uniformly throughout the analysis.

7. The study lacks any form of validation—either technical replicates or an independent validation cohort. The reproducibility of the sequencing data should be addressed, potentially through read saturation plots or duplicate library controls.

8. Figures are generally informative, but legends could be improved. For example, Figures 2 and 4 require clearer color legends and more descriptive axis labeling.

9. Figure 5 would benefit from including statistical annotations (e.g., asterisks for p-values) to indicate the significance of amino acid usage differences.

10. The manuscript briefly mentions clonal expansion and public clones but does not explore underlying mechanisms or implications. Are these changes driven by specific antigens? Is there evidence for immune exhaustion or senescence in MGUS/MM?

11. A more detailed discussion of how repertoire features may reflect tumor-immune interactions would be valuable.

12. The manuscript is generally well written but could benefit from some minor language polishing. Some expressions (e.g., “suggests a trend”) should be supported by statistical evidence or removed.

13. Gene and protein names should follow standard formatting conventions throughout.

Reviewer #2: I appreciate the authors’ comprehensive responses to the first-round comments. The revised version is improved in clarity, methodological rigor, and scientific presentation.

The study presents a valuable analysis of the TCRB repertoire in plasma cell dyscrasias, with integration of immune repertoire sequencing, machine learning-based classification, and antigen annotation. The distinction between global TCR diversity and cluster-level features appropriately highlights the complexity of T-cell biology in this disease context.

While I support the publication of this work, I have a few minor comments and suggestions as outlined below.

1. The MRD results are mentioned briefly. Does MRD status correlate with any TCR feature?

2. Line 259-261: The overlap analysis is interesting but lacks interpretation. Is this overlap more or less than expected by chance?

3. Line 266: How were variable importance scores interpreted across different models? Did top clusters in different models overlap significantly?

4. Please emphasize more clearly that differentially abundant clusters were identified despite no difference in total diversity, which strengthens the idea that qualitative rather than quantitative TCR features matter.

5. The limitation that many sequences did not map to known antigens may reflect gaps in current databases, not necessarily lack of disease relevance.

Reviewer #3: 1. Validation of Clustering Results

a. The current manuscript does not report how stable or robust the clustering results are. The authors didn’t clarify whether they ran the clustering multiple times to assess consistency.

b. No silhouette score, Davies–Bouldin index, or other clustering quality metric is reported to justify the choice of the number of clusters.

2. Sample Size and Overfitting

a. The ML models (especially clustering) are susceptible to overfitting, particularly with limited samples. It’s unclear whether the authors used methods to address this (e.g., bootstrapping, downsampling, or dimensionality selection tuning).

3. Biological Interpretation of Clusters

a. Some clusters are interpreted as being specific to disease state in their current manuscript, but this is not rigorously tested. A formal statistical test comparing cluster membership distributions across disease vs. healthy controls is needed.

b. Additionally, reporting effect sizes and confidence intervals would strengthen the inferences.

4. Lack of Integration with Clinical Data

a. Their study would benefit from correlating clusters or clonality features with available clinical parameters (e.g., disease stage, treatment status), if accessible.

5. Multiple Testing Correction

a. No indication is given that corrections for multiple hypothesis testing were applied, despite numerous comparisons. This should be addressed in their next round.

**Do you want your identity to be public for this peer review?** For information about this choice, including consent withdrawal, please see our Privacy Policy

Reviewer #1: No

Reviewer #2: No

Reviewer #3: No

---

## [Author Response · Author response to Decision Letter 2]

3 Sep 2025

Reviewer #1: This manuscript presents an analysis of the peripheral blood T-cell receptor (TCR) β-chain repertoire in patients with monoclonal gammopathy of undetermined significance (MGUS), multiple myeloma (MM), and healthy controls (HC), using high-throughput sequencing. The authors identify distinct TCR repertoire features across groups, including differences in clonal expansion, public/shared clones, and CDR3 amino acid usage.

The study offers a novel perspective on the immune microenvironment in plasma cell dyscrasias and highlights the potential utility of TCR repertoire profiling in disease stratification. However, there are several significant concerns related to cohort characterization, data interpretation, and methodological clarity that must be addressed before the manuscript can be considered for publication.

1. The manuscript lacks key clinical details about the MGUS and MM cohorts. It is not specified whether patients were treatment-naïve, or if any had received prior immunosuppressive therapy. Since TCR diversity is highly sensitive to immune status, such information is crucial for data interpretation.

We thank the reviewer for raising this important point. To clarify, the MGUS and MM cohorts include patients who were treatment naïve at the time of sample collection, and we have revised the methods section accordingly.

Revised Methods:

TCR sequencing was performed on the peripheral blood mononuclear cells (PBMCs) of 80 patients with untreated MGUS, 55 with SMM, and 31 with newly diagnosed MM before treatment (NDMM, S1 Table).

2. Additionally, information on infection or inflammatory status should be included, as these factors can influence TCR repertoire features independently of disease state.

We thank the reviewer for this suggestion. While infection and inflammatory status are key variables that can influence the TCR repertoire, this information was not systematically collected for our cohort per IRB protocol, and we are unable to include it in the analysis. We have now acknowledged this limitation in the discussion section of the manuscript.

Revised Discussion:

The biological significance of these differentially abundant clusters remains uncertain, but several possibilities may explain their enrichment in plasma cell dyscrasias. First, these clusters could represent tumor-reactive T cells that have undergone clonal expansion in response to malignant or pre-malignant plasma cell antigens. Second, they may reflect bystander T-cell expansions driven by chronic immune activation, microbial exposure, or tissue remodeling. Third, differences in the TCRB repertoire between patients and controls could be influenced by non-disease factors such as subclinical infections or vaccination history. We acknowledge that information regarding recent infections, inflammatory status, and prior immunization was not systematically collected for all study patients.

3. The study includes only 10 healthy controls, 8 MGUS, and 12 MM patients. The small cohort size limits the robustness and generalizability of the findings, especially for analyses of public clone distribution and amino acid bias.

We apologize if this was not sufficiently clear in the manuscript and would like to clarify the cohort numbers. The analysis was performed on 80 patients with untreated MGUS, 55 with SMM, 31 with newly diagnosed MM, and 166 healthy controls. To our knowledge, this is the largest T-cell repertoire study in patients with plasma cell dyscrasias to date.

4. The authors are encouraged to incorporate effect size metrics, confidence intervals, or bootstrapping to enhance statistical reliability.

We thank the reviewer for this constructive suggestion to enhance the statistical reliability of our findings. We agree that effect size metrics and confidence intervals are essential for robust interpretation and have incorporated these statistical measures throughout the manuscript:

• Confidence intervals: We have included 95% confidence intervals in our machine learning performance evaluation, specifically for the AUROC, accuracy, sensitivity, and specificity metrics, as displayed in Figure 3C.

• Effect size: For comparisons of amino acid characteristics and the Gini coefficient, effect size is determined by Wilcoxon rank-sum test p-values as shown in Figures 2 and 4.

• Bootstrapping: We employed a repeated stratified hold-out procedure (5 iterations of 80/20 splits) for internal validation. This bootstrapping-like approach allowed us to assess the stability and performance of our models on different subsets of the data, with the results summarized as means and 95% confidence intervals.

5. The criteria used to define “clonal expansion,” “shared clones,” and “public clones” are not clearly described. For example, what frequency threshold is used to identify expanded clones? Please provide a consistent and detailed definition of these terms in the Methods section and ensure they are applied uniformly throughout the analysis.

We thank the reviewer for highlighting the need to more clearly define these key terms. The terms in question are used in our Background and Discussion sections to provide context from the existing literature. To provide further clarity, below is how we interpret these concepts:

• Clonal expansion: In the cited literature, this often refers to an increase in the frequency of specific T-cell clones, sometimes identified by an arbitrary frequency threshold. In contrast, our study avoids such thresholds by measuring the repertoire clonality as a continuous variable using the Gini coefficient.

• Shared and public clones: The literature often uses “shared clones” to refer to identical TCR sequences found in different individuals, while “public clones” are well-known examples of these shared sequences, such as the ones listed in the databases we queried. Our study moves beyond simple sequence identity to identify functionally shared immune responses using ClusTCR to cluster structurally similar TCRs.

7. The study lacks any form of validation—either technical replicates or an independent validation cohort. The reproducibility of the sequencing data should be addressed, potentially through read saturation plots or duplicate library controls.

We thank the reviewer for raising the critical point of study validation and reproducibility. We acknowledge that validating our findings in a separate, independent patient cohort would be the gold standard. While this was not feasible for the present study, we did incorporate a rigorous validation step for our primary machine learning findings. Specifically, we used a stratified 70/30 split of our data. The 30% held-out test set was completely isolated from all feature selection and model training steps. This held-out set functions as an independent validation set for the machine learning model, and the performance on this set (AUROC 0.859) demonstrates the generalizability of the identified TCRB signature.

8. Figures are generally informative, but legends could be improved. For example, Figures 2 and 4 require clearer color legends and more descriptive axis labeling.

We thank the reviewer for their thoughtful feedback on our figures and for the suggestion to improve clarity. We have used distinct colors for the box plots in Figures 2A-D and 4B to provide clear visual separation between the different experimental groups and timepoints. We find this is particularly helpful for the reader in quickly distinguishing the adjacent plots within the dense, multi-panel figures, thereby improving overall readability. All other figures have legends describing the meaning of the color.

9. Figure 5 would benefit from including statistical annotations (e.g., asterisks for p-values) to indicate the significance of amino acid usage differences.

We thank the reviewer for the suggestion to enhance the presentation of our statistical findings. We believe this comment refers to Figure 4B, which details the differences in physicochemical properties, as this is where the significance of amino acid characteristics is presented. In designing the figure, we made the choice to present the exact p-values rather than using asterisk annotations on the plot. Our rationale is that providing the precise p-value offers a higher degree of statistical detail and transparency than the categorical thresholds represented by asterisks

10. The manuscript briefly mentions clonal expansion and public clones but does not explore underlying mechanisms or implications. Are these changes driven by specific antigens? Is there evidence for immune exhaustion or senescence in MGUS/MM?

We thank the reviewer for these excellent questions. In our analysis, we did not observe clonal expansion of public clones, nor a disproportionate abundance of public clones when comparing healthy individuals to those with plasma cell dyscrasias. In the discussion, we note that “future studies should focus on identifying the antigenic targets of the TCRs that comprise the differentially abundant TCRB clusters identified in this study. This could provide valuable insights into the mechanisms underlying the immune response in plasma cell dyscrasias and potentially reveal novel therapeutic targets. Additionally, exploring the phenotypic states and functional properties of these distinct T-cell populations could further elucidate their role in disease pathogenesis and response to therapy.”

11. A more detailed discussion of how repertoire features may reflect tumor-immune interactions would be valuable.

We thank the reviewer for these insightful comments, which address key areas of interest in the field. Our study was focused on the TCR repertoire and was not designed to directly assess antigen specificity or T-cell functional states. However, our findings provide a foundation for these critical future investigations. In the discussion, we acknowledge that “The biological significance of these differentially abundant clusters remains uncertain, but several possibilities may explain their enrichment in plasma cell dyscrasias. First, these clusters could represent tumor-reactive T cells that have undergone clonal expansion in response to malignant or pre-malignant plasma cell antigens. Second, they may reflect bystander T-cell expansions driven by chronic immune activation, microbial exposure, or tissue remodeling. Third, differences in the TCRB repertoire between patients and controls could be influenced by non-disease factors such as subclinical infections or vaccination history.”

12. The manuscript is generally well written but could benefit from some minor language polishing. Some expressions (e.g., “suggests a trend”) should be supported by statistical evidence or removed.

We thank the reviewer for their positive feedback and for the helpful suggestion to ensure our language is precise. Our use of the term “trend” is intended to describe a consistent pattern in the data that we believe may be biologically relevant, even when it does not meet our pre-defined threshold for statistical significance. To ensure full transparency, in every instance where we mention a trend (e.g., in our discussion of Figures 2 and S4), we have also reported the outcome of the corresponding statistical test. This allows the reader to independently assess both the visual pattern and its level of statistical support.

13. Gene and protein names should follow standard formatting conventions throughout.

We have carefully reviewed the entire manuscript to ensure that all gene and protein names adhere to standard formatting conventions.

Reviewer #2: I appreciate the authors’ comprehensive responses to the first-round comments. The revised version is improved in clarity, methodological rigor, and scientific presentation.

The study presents a valuable analysis of the TCRB repertoire in plasma cell dyscrasias, with integration of immune repertoire sequencing, machine learning-based classification, and antigen annotation. The distinction between global TCR diversity and cluster-level features appropriately highlights the complexity of T-cell biology in this disease context.

While I support the publication of this work, I have a few minor comments and suggestions as outlined below.

1. The MRD results are mentioned briefly. Does MRD status correlate with any TCR feature?

We thank the reviewer for this question. We investigated whether achieving sustained MRD negativity in high-risk SMM patients was associated with overall TCRB diversity. As shown in Figure 2E and stated in the Results and Discussion sections, our analysis did not reveal a statistically significant difference in TCRB diversity between patients who achieved sustained MRD negativity and those who did not.

2. Line 259-261: The overlap analysis is interesting but lacks interpretation. Is this overlap more or less than expected by chance?

We thank the reviewer for raising this important point. In our initial analysis, the Venn diagram in Figure 3A included all available samples, which meant that some patients contributed more than others if they had multiple time points. To avoid potential bias from unequal sampling and to ensure that each patient contributed equally, we revised the analysis to include only baseline samples. This approach provides a clearer and more consistent representation of exclusivity across disease groups.

To assess whether the observed group-specific exclusivity differed from what would be expected by chance, we performed a permutation test. Specifically, we preserved the number of baseline samples in each disease category (Healthy, MGUS, SMM, MM) and randomly permuted group labels across samples 1,000 times. For each permutation, we recalculated the proportion of clusters exclusive to each group. We then compared the observed exclusivity to the null distribution, computing mean and two-sided permutation p-values.

The results below indicate that the observed exclusivity percentages were generally consistent with chance expectations. MGUS (10.7% observed vs. 10.0% expected, p = 0.42) and MM (3.4% vs. 3.6%, p = 0.77) fell within the permutation-derived confidence intervals, while SMM (8.6% vs. 6.6%, p = 0.016) and healthy (21.3% vs. 24.4%, p = 0.04) showed modest but statistically significant enrichment. Overall, these results suggest that the exclusivity of TCR clusters across diagnostic categories is largely consistent with random expectation, with SMM and healthy representing potential exceptions. The methods and results sections have been updated accordingly.

[See uploaded response letter with figure showing bar plot comparing observed vs expected exclusive cluster size.]

Revised Methods:

Group-specific TCRB cluster exclusivity was evaluated by permutation testing, in which baseline sample labels were shuffled 1,000 times while preserving group sizes, and observed exclusivity was compared against the resulting null distribution to derive empirical p-values.

Revised Results:

Overlap analyses between baseline sample clusters demonstrated that 21.3% are exclusive to healthy, 10.7% to MGUS, 8.6% to SMM, and 3.4% to MM (Figure 3A). A permutation test confirmed that the observed exclusivity percentages were consistent with chance expectations for MGUS (10.7% observed vs. 10.0% expected, p = 0.42) and MM (3.4% vs. 3.6%, p = 0.77) while SMM (8.6% vs. 6.6%, p = 0.016) and healthy (21.3% vs. 24.4%, p = 0.04) showed evidence of enrichment beyond random expectations.

3. Line 266: How were variable importance scores interpreted across different models? Did top clusters in different models overlap significantly?

We thank the reviewer for this excellent suggestion. We compared variable importance across the four top-performing models (Random Forest, Support Vector Machines with RBF kernel, Elastic Net, and Neural Network) using pairwise Spearman correlations of the variable-importance rankings. This analysis showed that Random Forest and Support Vector Machines demonstrated the greatest similarity (r ≈ 0.47), while Random Forest also exhibited modest agreement with Elastic Net (r ≈ 0.23) and Neural Network (r ≈ 0.15). In contrast, Elastic Net showed little to no alignment with either Support Vector Machines (r ≈ –0.18) or Neural Network (r ≈ 0.02). These results indicate that

---

## [Decision Letter · Decision Letter 2]

22 Sep 2025

Machine learning reveals distinct T-cell receptor clusters in plasma cell dyscrasias compared to healthy controls

PONE-D-25-19122R2

Dear Dr. Coffey,

We’re pleased to inform you that your manuscript has been judged scientifically suitable for publication and will be formally accepted for publication once it meets all outstanding technical requirements.

Kind regards,

Jian Wu, M.D, Ph.D

Academic Editor

PLOS ONE

Additional Editor Comments (optional):

The author addressed all the comments and the manuscript was much improved.

Reviewers' comments:

Reviewer's Responses to Questions

**Comments to the Author**

Reviewer #2: All comments have been addressed

Reviewer #3: All comments have been addressed

2. Is the manuscript technically sound, and do the data support the conclusions?

Reviewer #2: Yes

Reviewer #3: Yes

3. Has the statistical analysis been performed appropriately and rigorously?

Reviewer #2: Yes

Reviewer #3: Yes

4. Have the authors made all data underlying the findings in their manuscript fully available?

Reviewer #2: Yes

Reviewer #3: Yes

5. Is the manuscript presented in an intelligible fashion and written in standard English?

Reviewer #2: Yes

Reviewer #3: Yes

Reviewer #2: The authors have fully addressed all prior reviewer concerns. The revised manuscript is clear, rigorous, and reproducible, with well-presented figures and transparent methodology. I find no outstanding issues. So, I recommend acceptance in its current form.

Reviewer #3: All previous statistical concerns have been properly addressed by authors properly. No further questions.

**Do you want your identity to be public for this peer review?** For information about this choice, including consent withdrawal, please see our Privacy Policy

Reviewer #2: No

Reviewer #3: No

---

## [Editor Report · Acceptance letter]

PONE-D-25-19122R2

PLOS ONE

Dear Dr. Coffey,

I'm pleased to inform you that your manuscript has been deemed suitable for publication in PLOS ONE. Congratulations! Your manuscript is now being handed over to our production team.

Kind regards,

on behalf of

Dr. Jian Wu

Academic Editor

PLOS ONE